# A Novel Method for Adjusting the Taper and Adaption of Automatic Tooth Preparations with a High-Power Femtosecond Laser

**DOI:** 10.3390/jcm10153389

**Published:** 2021-07-30

**Authors:** Fusong Yuan, Shanshan Liang, Peijun Lyu

**Affiliations:** 1Center of Digital Dentistry, Peking University School and Hospital of Stomatology, Beijing 100081, China; yuanfusong@bjmu.edu.cn; 2Department of Prosthodontics, Peking University School and Hospital of Stomatology, Beijing 100081, China; 3National Engineering Laboratory for Digital and Material Technology of Stomatology, Beijing 100081, China; 4NHC Key Laboratory of Digital Technology of Stomatology, Beijing 100081, China; 5Beijing Key Laboratory of Digital Stomatology, Beijing 100081, China; 6National Clinical Research Center for Oral Diseases, Beijing 100871, China; 7Second Clinical Division, Peking University Hospital of Stomatology, Beijing 100081, China; Liangshanshan@pkuss.bjmu.edu.cn

**Keywords:** adaption, laser light-off delay, taper, tooth preparation robot

## Abstract

This study explored the effect of the light-off delay setting in a robotically controlled femtosecond laser on the taper and adaption of resin tooth preparations. Thirty resin teeth (divided into six equal groups) were studied under different light-off delay conditions. Tapers from six vertical sections of the teeth were measured and compared among the light-off delay groups. The mean taper decreased from 39.268° ± 4.530° to 25.393° ± 5.496° as the light-off delay increased (*p* < 0.05). The average distance between the occlusal surfaces of the scanned data and the predesigned preparation data decreased from 0.089 ± 0.005 to 0.013 ± 0.030 μm as the light-off delay increased (*p* < 0.05). The light-off delay of the femtosecond laser is correlated with the taper and adaption of automatic tooth preparations; this setting needs to be considered during automatic tooth preparation.

## 1. Introduction

Clinical tooth preparation currently relies on the use of an air turbine handpiece or commercial dental lasers. However, these two methods have limitations, and their effectiveness relies greatly on the experience of the dentist. Misuse of these tools can result in excessive or inadequate preparation and potential injury to the gingiva, tongue, and buccal and labial mucosa. Additionally, high-speed dental turbines generate vibrations and sharp noises, which can make patients uncomfortable [1,2,3,4,5]. Furthermore, the currently used laser and dental turbines may stimulate the dental pulp, which can lead patients to endure varying degrees of pain [6,7,8,9]. Taper is one of the check points for achieving high-quality tooth preparation. It refers to the angle of aggregation of the tooth from the level of occlusion to the level of the gingiva after tooth preparation. The clinical standard for taper is 2–6°. However, due to differences in dentist proficiency and the difficulty of working in the narrow oral space with a limited visual field, it is often difficult to meet this requirement. An experiment by Aminian et al. showed that when preparing teeth on models, dentists often performed insufficient labial reduction and excessive incisional reduction [10]. Leempoel et al. found that the minimum average taper of a tooth preparation ranged from 7.75° to 15.15° [11]. Chen et al. showed that when the taper increased from 6° to 10°, cast retention decreased markedly, as did the success rate [12].

To overcome the drawbacks of manual tooth preparation techniques and to achieve an automatic ablation process, Yuan et al. utilized a three-axis, numerically controlled picosecond laser to prepare teeth successfully. Their findings validated a laser-based automatic tooth preparation technique (Figure 1). The robot contained (1) an intraoral scanner to obtain three-dimensional (3D) data of the target tooth and tooth fixture; (2) computer-aided design (CAD)/computer-aided manufacturing (CAM) software for generating the ablation path of the laser; (3) an effective low-heat ultrashort pulse-laser generator; (4) a six-degrees-of-freedom light-guiding arm; (5) a robotic system comprising two high-speed galvanometers and one focusing lens for tooth preparation; and (6) a tooth fixture that connected the robotic device to the target tooth [13].

Lasers have been used in the field of stomatology for a long time, with applications in oral mucosal disease, periodontal disease, dental caries, and peri-implantitis. In particular, it has a remarkable role in oral soft-tissue diseases. In recent years, due to the high precision of ablation and few thermal side effects [14,15,16,17,18,19], ultrashort pulse lasers have been widely used. According to previous studies, the hard tissue of teeth can be ablated at a low energy density by a femtosecond laser; the thresholds of the enamel and dentin were 0.6–2.2 and 0.3–1.4 J/cm^2^, respectively [20,21,22,23]. Sun et al. studied the efficacy of a titanium–sapphire femtosecond laser system (CPA-2001, Clark-MXR, Dexter, MI, USA) with respect to ablation efficiency and surface morphology of the dental hard tissues. The results showed that the surface of the hard tissue was smooth, and the boundary was clear after femtosecond laser ablation [24]. In previous studies, a femtosecond laser has been used to ablate a polylactic acid tubular product, producing a 20-μm-wide slit with a smooth inner wall and no residue or edge trimming [25,26]. Moreover, Wang et al. used femtosecond lasers to process 50-μm-thick poly(lactic-co-glycolic acid) to obtain microvascular stents [27]. Thus, the femtosecond laser is expected to become a tool for high-precision tooth preparation.

However, current research on the use of lasers in dentistry has mainly focused on the microscopic morphology, thermal effect, and ablation efficiency of lasers on the dental hard tissues [14,28,29,30,31]. There have been few studies on the accuracy of robotically controlled femtosecond laser ablation in tooth preparation. In a preliminary experiment, we have shown that when the light-off delay changed, the taper of the tooth preparation changed correspondingly. The light-off delay is a parameter of tooth preparation robotic control software. It refers to the time the laser beam stays on before the laser light is turned off. We hypothesized that if the laser light cannot be turned off immediately at the end of the laser-marking process, the marking will be too light; however, if the laser light-off delay is too long, the marking will be too heavy, which could then affect the taper.

The objective of the present study was to test our hypothesis that the light-off delay affects the results of tooth preparation.

## 2. Materials and Methods

Automatic tooth preparation was performed using a robot that controlled a femtosecond laser at 1030 nm, with a pulse width of <300 fs, a repetition rate of 200 kHz, an average output power of 6 W, a laser spot diameter of 80 μm, and a scan rate of 2580 mm/s. The TANGERINE femtosecond laser system (Amplitude Systemes, Bordeaux, France) was used.

Thirty standard composite resin teeth (molars) (A5SAN-500; Nissin Dental Products Inc., Kyoto, Japan) were randomly divided into six equal groups and were examined under different light-off delay conditions. In Groups 1–6, the period of light-off delay varied from 180 to 480 ms, with a 40-ms interval between groups, while the other parameters remained constant.

### 2.1. Data Acquisition

Resin teeth were fixed onto a dental model, and 3D data of the model with (Data I) and without (Data II) tooth fixture were obtained using an oral scanner. Then, Data I and Data II were registered into one coordinate by a common area registration method and the target tooth data were extracted using Geomagic Studio (Geomagic Studio 2013; Raindrop Geomagic, Research Triangle, NC, USA). The design of the tooth preparation was completed according to the predesigned preparation data of the full metal crown designed by self-developed CAD/CAM software. A 28-μm preparation depth and 400 total preparation layers were adopted to complete the discrete layered sections of the 3D CAD model of the prepared teeth. Scanning line filling and scanning path planning were performed according to each section’s outline information.

### 2.2. Tooth Preparation

Tooth preparation was automated with a femtosecond laser controlled by a dental robot (Figure 2) with a single depth of ablation at 28 μm. Preparation of the tooth surface was performed by the robot controlling the femtosecond laser to obtain a 2D-plane image of a specific layer (Figure 3). This process was conducted using high-speed 2D galvanometers. Five layers of pulse scanning and a 28-μm single-layer depth were prepared on the tooth surface; thus, the focusing lens moved forward by 28 μm. Each step was followed by another round of sectioning and scanning preparation. This process continued until a 3D virtual model of the tooth preparation was generated.

### 2.3. Software Measurement and Statistical Analysis

The 30 tooth preparations were digitally scanned with a 3D laser scanner (IScan D104i, Imetric 3D SA, Courgenay, Switzerland). Scanned data were converted to a stereolithography file. The 3D CAD software (Geomagic Studio 2013) was used to generate 3D reconstructions of these data (Figure 4). The tapers of six virtually vertical planes of each tooth preparation were measured using Geomagic Qualify software (Geomagic Qualify 2013) (Figure 5). Taper was determined by analyzing the taper between two opposing surfaces of all six planes. The mean values for each group were calculated, and a total of 30 tapers were measured for each light-off delay group. The predesigned preparation data and scanned data were input into the Geomagic Studio software. Eight different landmarks were selected from the predesigned data; these same landmarks were identified on scanned data and were used for manual registration. Then, the common regional register module was used to register the two sets of data. A bias module was subsequently used to measure the average distance (Figure 6). Five average distances were measured for each group. The overall average error between the predesigned data and scanned data was analyzed using the rank sum test in SPSS 19.0 (IBM SPSS Inc., Chicago, IL, USA) to compare the statistical significance of differences between the mean of the tapers and the average distance among groups.

## 3. Results

Automatic preparation of the resin teeth using a tooth preparation robot is shown in Figure 7.

The mean taper measurements for Groups 1–6 are shown in Table 1 and Figure 8; the taper decreased as the light-off delay increased. The mean taper decreased from 39.268° ± 4.530° to 25.393° ± 5.496° as the light-off delay increased. In this regard, there were significant differences between Groups 1 and 3, Groups 1 and 4, Groups 1 and 5, Groups 1 and 6, Groups 2 and 3, Groups 2 and 4, Groups 2 and 5, Groups 2 and 6, Groups 3 and 6, Groups 4 and 6, and Groups 5 and 6 (*p*-value < 0.05). In contrast, there were no significant differences between Groups 1 and 2, Groups 3 and 4, and Groups 4 and 5 (*p*-value > 0.05) (Table 2).

Table 3 and Figure 9 showed the comparison of the average distance between the scanned data and predesigned preparation data among Groups 1–6; these values decreased as the light-off delay increased; the average distance decreased from 0.089 ± 0.005 to 0.013 ± 0.030 μm as the light-off delay increased. In this regard, there were significant differences between Group 1 and the other groups, Group 2 and the other groups, and between Groups 3 and 5, Groups 3 and 6, Groups 4 and 5, and Groups 4 and 6 (*p*-value < 0.05); in contrast, no significant differences were observed between Groups 3 and 4 and between Groups 5 and 6 (*p*-value > 0.05) (Table 4).

## 4. Discussion

The convergence degree and the amount of prepared tooth are two important indicators of tooth preparation. This study explored a new method to improve the precise control of the convergence degree and the amount of prepared tooth and further enhance the accuracy of automated tooth preparation using a femtosecond laser. The automated tooth preparation system can help dentists improve the accuracy of tooth preparation, reduce dentists’ fatigue strength, and improve the patient experience.

In this experiment, we performed automatic tooth preparation using robotic control of a femtosecond laser for automatic 3D ablation of the target crown. Use of different light-off delay settings for the laser affected both the taper and distance between the scanned data and predesigned preparation data, with a decrease in these parameters as the light-off delay increased. However, a delay of 480 ms led to carbonization in the lower parts of the tooth preparation.

According to previous research, ideal 3D ablation should ensure that the thickness of a single slice is equal to the actual ablation depth of each laser layer. Therefore, obtaining a suitable single-depth ablation is a prerequisite for ensuring high-precision, high-efficiency, and high-quality 3D laser ablation. Yuan et al. conducted a preliminary study on the method of controlling the dentin depth error of a computerized, numerically controlled picosecond laser ablation. They found that the normal step size is positively correlated with the error of the ablation depth and the total depth-of-ablation error [32]. In the present study, we performed a single ablation of three layers using a single-layer ablation depth of 28 μm to reduce the error caused by multilayer ablation. This allowed us to explore the precision of this resin tooth preparation method and evaluate its effects.

In view of the anisotropy of the natural tooth structure, the irregular shape of the crown, and the different abilities of the laser to ablate enamel and dentin, resin teeth with the same shape as a natural tooth were used. Such a homogeneous sample can better reflect the effects of different laser parameters on the taper and adaption of the tooth preparation. The power and parameters used in this experiment could also be extended to natural teeth.

Laser energy exhibits a non-uniform Gaussian distribution. After the resin surface leaves the focal plane, the energy density at the periphery of the spot decreases, and the central energy density may still be higher than the target material’s ablation threshold. Therefore, the single-spot ablation surface may appear crater-like, and the taper of the laser ablation cannot meet the standard of the predesigned data. In the actual ablation process, the sidewall of each ablation layer contacting the resin material showed a reduced ablation phenomenon. Since ablation is accomplished through a layer-by-layer accumulative process, the accumulation of reduced ablation causes the tilt angle of the sidewall to increase. Consequently, the taper of the preparation body also increases, and this is not conducive to controlling the taper of the tooth preparation. In this study, we showed that when the light-off delay increased, taper decreased. In other words, the light-off delay could effectively reduce the taper of the preparation. However, the light-off delay cannot be increased indefinitely because a greater laser dwell time leads to heat accumulation and eventually carbonization of the target material.

The powerful peak power of the femtosecond laser enables the target tissue to form plasma directly [33,34], and the laser energy is used exclusively for material ablation, with the ablation rate increasing linearly with the energy density [35]. However, the actual ablation process has complex effects [36]. The arrival of the first laser pulse turns the surface tissue into plasma. If the formed plasma cannot be dissipated in time, it will form a barrier that absorbs or reflects the energy of the next laser pulse, preventing this laser energy from reaching the surface of the material. Excessive laser frequency, energy density, or a low scan rate can cause plasma buildup and a plasma-shielding effect [19]. The appearance of white powder on the resin tooth surface suggests that the ablated material had not entirely been converted to plasma. Thus, the increase in taper may be related to the incomplete formation of plasma, and these correlation parameters should be explored.

The larger tapers of the resin tooth preparation may also be a result of the poor position repeatability of the laser beam. The position repeatability of the laser beam used in this study was ±0.04 mm according to a preliminary experiment. This error may be one of the reasons for the increase in taper. Therefore, improved results may be obtained by using a robot with better position repeatability in future.

The average distance between the scanned data and predesigned data is an important measurement for the adaption of full-crown restoration. A large average distance indicates low adaption. During the registration process, the Geomagic Studio software calculated 3D suitability according to each point cloud; i.e., the software automatically calculated the paired root mean square for quantitative analysis. The magnitude of the difference can be specifically reflected in the positive and negative values on the deviation chromatogram for qualitative analysis. In the deviation chromatogram, the surface of the predesigned data was used as the origin. A positive value represents the distance away from the surface of the predesigned data, and a negative value reflects the distance from the surface of the predesigned data in the negative direction. This method calculates all paired point clouds for quantitative analysis, thereby obtaining more suitable information and avoiding the problem of data loss due to human handling of experimental objects [37]. In the ablation process, the current ablation plane and focal plane of the laser were in the same position (in the normal direction), so that the entire surface of the resin was irradiated by each spot, which reached the same or minimally different depths of ablation in a single ablation layer. This reduces the ablation error, resulting in more accurate depth control and a preferable adaption of the occlusal surface. However, the average distance that measures the deviation between the scanned data and predesigned data includes errors in scanning, data registration, software analysis, and laser ablation. This includes deviation of the occlusal and axial surfaces, in which the average distance of the occlusal surface is small and steady for the aforementioned reasons. When the light-off delay increased, the average distance of the axial surface decreased, similar to taper. Consequently, adaption also decreased.

This study has some limitations. In the actual application process, the effect of light-off delay on the taper would change according to the different materials used. One of the limitations of this study was that there is a certain difference between the tapers of tooth preparation and the tapering of clinical requirements. This might be solved by adjusting the taper of the model and exploring the relationship between other parameters and taper. Another limitation of this experiment was the use of resin teeth to explore the effect of the light-off delay on tooth preparation. During clinical tooth preparation, the dental hard tissues that need to be removed often contain both enamel and dentin. Using the same femtosecond laser parameters for materials with different ablation capabilities, the effect of the light-off delay on the taper and average distance of the teeth needs to be explored in the future. Nevertheless, our experiment revealed a clear trend.

## 5. Conclusions

We showed that the light-off delay setting of the femtosecond laser influences both taper and adaption in automatic tooth preparations. Thus, this setting needs to be considered during automatic tooth preparation.

## Figures and Tables

**Figure 1 jcm-10-03389-f001:**
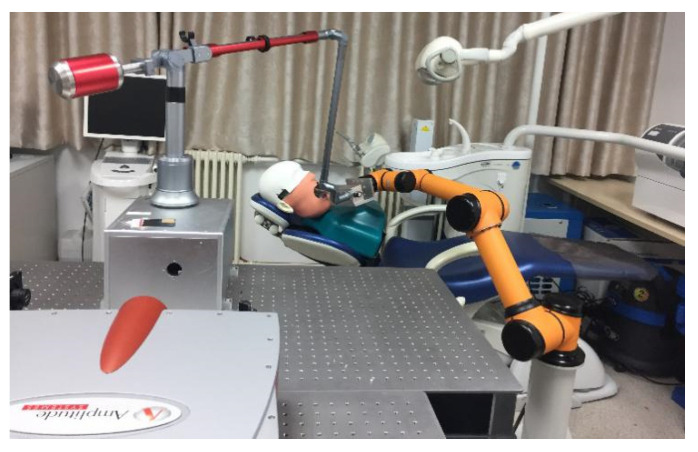
Automatic tooth preparation robot.

**Figure 2 jcm-10-03389-f002:**
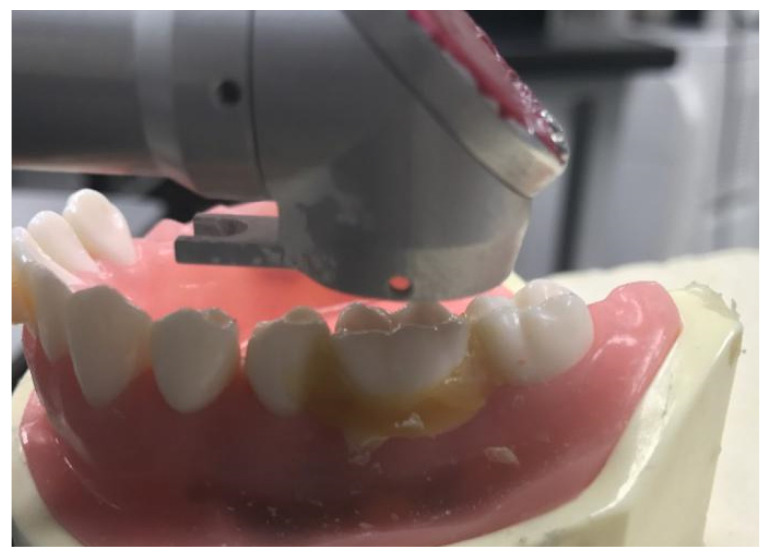
Resin tooth ablated automatically by the dental robot.

**Figure 3 jcm-10-03389-f003:**
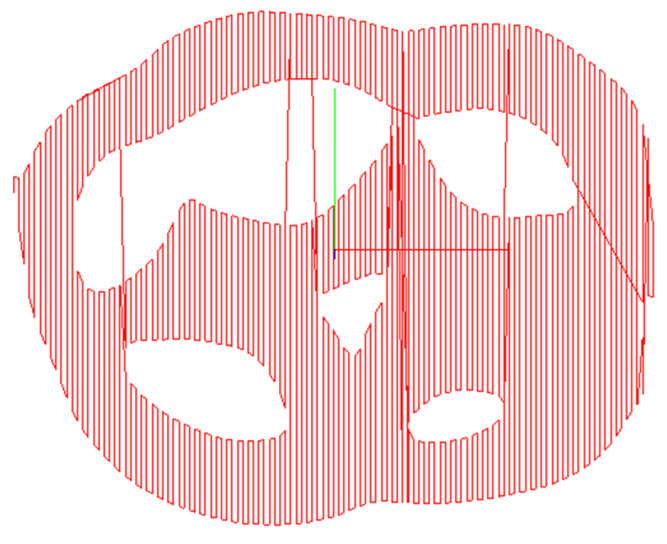
Design of the path for laser scanning for single-layered preparation.

**Figure 4 jcm-10-03389-f004:**
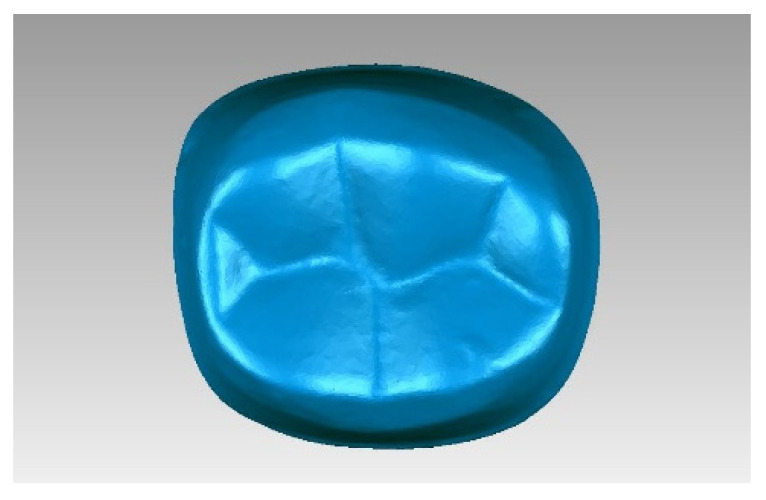
Stereolithography file of the tooth preparation.

**Figure 5 jcm-10-03389-f005:**
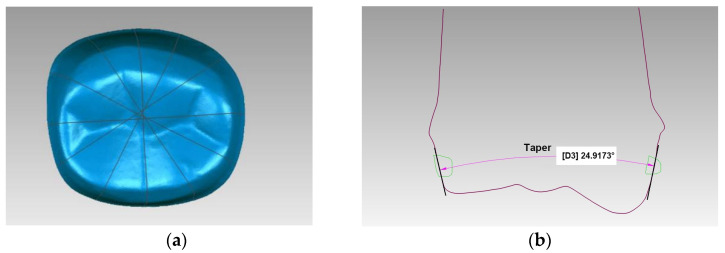
Six vertical planes of the tooth preparation (**a**); measurement of the taper of a resin tooth preparation in Geomagic Qualify (**b**).

**Figure 6 jcm-10-03389-f006:**
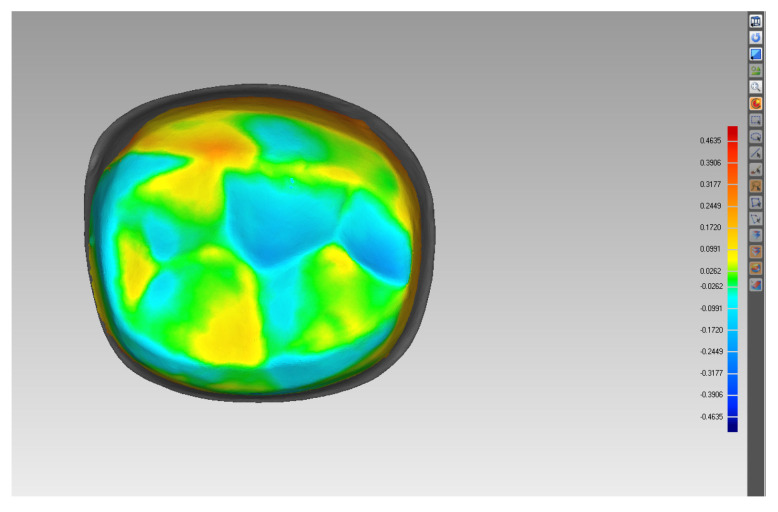
Measurement of the average distance between the scanned data and predesigned preparation data.

**Figure 7 jcm-10-03389-f007:**
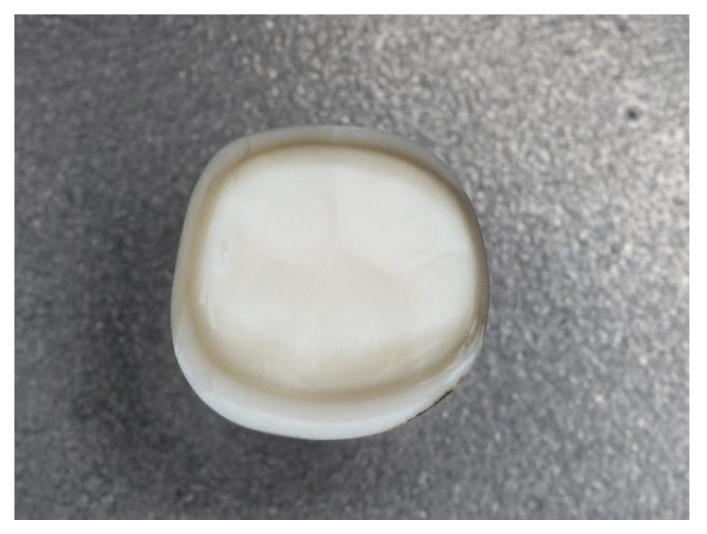
Resin tooth preparation.

**Figure 8 jcm-10-03389-f008:**
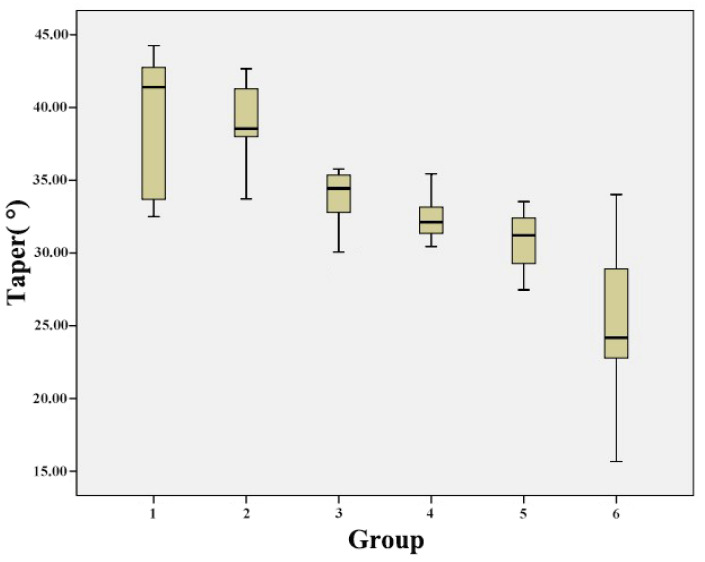
Influence of light-off delay on taper in the different groups.

**Figure 9 jcm-10-03389-f009:**
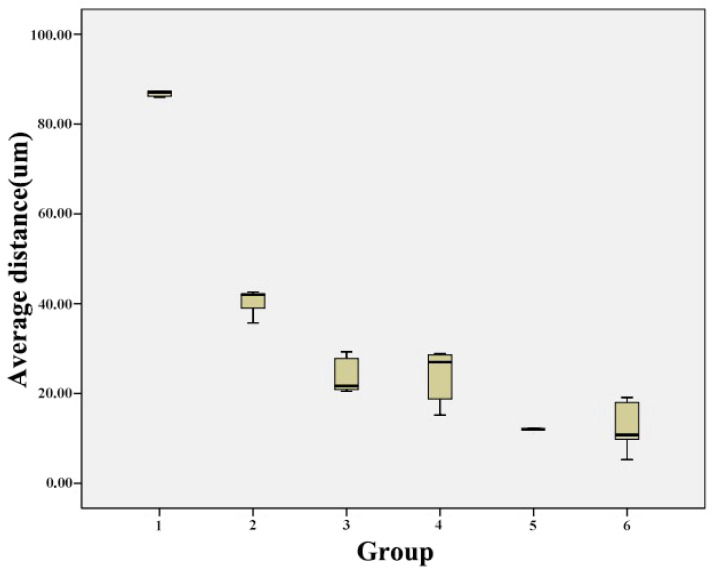
Influence of light-off delay on the average distance between the scanned data and predesigned preparation data.

**Table 1 jcm-10-03389-t001:** Descriptive statistics of the taper (in °) of the tooth preparations made using different light-off delay settings.

Group	Light-Off Delay (ms)	Mean (°)	SD (°)
1	180	39.268	4.530
2	240	38.806	2.888
3	300	33.652	2.257
4	360	32.476	1.553
5	420	31.159	2.078
6	480	25.393	5.496

SD, standard deviation.

**Table 2 jcm-10-03389-t002:** Statistical analysis (*p*-value) of the taper (in °) of the tooth preparations made using different light-off delay settings.

Group	2	3	4	5	6
1	0.749	0.000	0.000	0.000	0.000
2	-	0.001	0.000	0.000	0.000
3	-	-	0.416	0.087	0.000
4	-	-	-	0.363	0.000
5	-	-	-	-	0.000

**Table 3 jcm-10-03389-t003:** Descriptive statistics of the average distance (in μm) between the scanned data and predesigned data of the tooth preparation with different light-off delays.

Group	Light-off Delay (ms)	Mean (μm)	SD (μm)
1	180	0.089	0.005
2	240	0.040	0.003
3	300	0.024	0.042
4	360	0.024	0.006
5	420	0.013	0.025
6	480	0.013	0.030

SD, standard deviation.

**Table 4 jcm-10-03389-t004:** Statistical analysis (*p*-value) of the average distance (in μm) between the scanned data and predesigned data of the tooth preparation with different light-off delays.

Group	2	3	4	5	6
1	0.000	0.000	0.000	0.000	0.000
2	-	0.000	0.000	0.000	0.000
3	-	-	0.911	0.001	0.001
4	-	-	-	0.001	0.001
5	-	-	-	-	0.932

## Data Availability

The data presented in this study are available on request from the corresponding author.

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
