# Peer review of "A Novel Method for Adjusting the Taper and Adaption of Automatic Tooth Preparations with a High-Power Femtosecond Laser"

_jcm, 2021, doi:10.3390/jcm10153389_

Round 1

Reviewer 1 Report

The authors investigated “A Novel Method for Adjusting Taper and Adaption of Automatic Tooth Preparation with a High-Power Femtosecond Laser“. In particular, the effect of different light-off conditions were examined.

Introduction is well written and provides an overview with outlining significance and aim of the study. Hypothesis is clearly outlined.

Materials and Methods
In this study, 30 composite resin teeth were divided in 6 groups with different period of light-off delay and prepared by a robotic controlled femtosecond laser system.

Results

Main result is correlation between increase of light-off delay with decrease of taper.

Discussion

This section is very well written by the authors with up to date references. The shortcomings of the study are even listed and discussed adequately. English language does not need corrections.

Author Response

Thank you for your comments and your hard work.

Reviewer 2 Report

The context is the use of femtosecond laser for tooth preparation using a robotic system.

The objective of the study is very specific: it relates to the impact of the light-off delay of the laser on the taper obtained at the end of the tooth preparation.

The taper obtained in the studied varied between 25° to 39°: in the introduction, you report that the expected taper for tooth preparation ranges between 7° to 15° (Leempoel et al., ref 9, page 1 line 40): in this regard, it seems that the obtained taper is too high??

The p value is not expressed in the dedicated column of Table 1 and Table 2: you should remove this column from these tables and add another table to show the p value results.

Table 1 / Figure 8 and Table 2 / Figure 9 are redundant. What is the added value of showing the results in 2 different forms? I suggest to choose between figures 8-9 OR Table 1-2.

You are using resin teeth in this study: have you any clue to extrapolate your results on actual teeth? Moreover, you can expect different cutting behavior between enamel and dentin? Also, how do you plan to manage laser power when these 2 tissues will co-exist in a same slice?

Reviewer 3 Report

The topic of the manuscript is the novel method of automatic tooth preparation.

The abstract and the main text of the article are informative. The Introduction clearly presents the laser technique for tooth preparation using robots. The section “Material and Methods” precisely explains the chosen study design. The section “Results” should be improved. The Discussion is well written, including the limitations of the study. The Conclusions are general “take-home” messages.

Some following points must be clarified/corrected for the further processing of this article.

Merits-related comments:

  1. In Tables 1 and 2, the p-value column is completely unclear and incompletely presented. Therefore, p-values between each pair of groups (1 vs. 2, 1 vs. 3, etc.) should be presented in a separate table as a matrix.
  2. Additionally, it is recommended to add more comments in the text on the results presented in the Figures and Tables to make them easier to understand.
  3. It would be useful to enrich the Discussion in the clinical comments to demonstrate future practical implications for the work of dentists.
  1. Also, it is suggested to add more recent articles from 2017-2021 to the references in the Discussion and the Introduction.

Technical comments:

  1. In Figures 8 and 9 the groups should be numbered as 1, 2, 3, 4, 5, 6 (not 1.00, 2.00 etc.). Also, numbered outliers and extreme points should rather be removed.
  2. In Figure 9 on the y-axis should be the unit micrometre as in Table 2 (not millimetre).
  3. The citation list could be corrected. References should contain doi (if available).

Round 2

Reviewer 3 Report

The authors have partially improved the manuscript. Nevertheless a few technical comments on the statistical analysis:

  1. In all tables it would be better to approximate values to thousandths.
  2. The new tables are underwhelming. I propose (as before) a simplification including p-values as a matrix, possibly mean difference values in the additional lines (but this can be calculated from earlier tables, so is not necessary). Sample table below:

2

3

4

5

6

1

2

-

3

-

-

4

-

-

-

5

-

-

-

-

  1. Instead of P it should be p-value.
  2. Although another Reviewer suggested removing the graphs, I would consider improving and restoring them for better visualization of the dependencies - leaving or removing them to the Editor's discretion.
